# Blinatumomab as a Bridge Therapy for Hematopoietic Stem Cell Transplantation in Pediatric Refractory/Relapsed Acute Lymphoblastic Leukemia

**DOI:** 10.3390/cancers14020458

**Published:** 2022-01-17

**Authors:** Katarzyna Pawinska-Wasikowska, Aleksandra Wieczorek, Walentyna Balwierz, Karolina Bukowska-Strakova, Marta Surman, Szymon Skoczen

**Affiliations:** 1Department of Pediatric Oncology and Hematology, Institute of Pediatrics, Jagiellonian University Medical College, 31-663 Krakow, Poland; walentyna.balwierz@uj.edu.pl (W.B.); szymon.skoczen@uj.edu.pl (S.S.); 2Department of Pediatric Oncology and Hematology, University Children’s Hospital, 30-663 Krakow, Poland; 3Department of Clinical Immunology, Institute of Pediatrics, Jagiellonian University Medical College, 31-663 Krakow, Poland; k.bukowska-strakova@uj.edu.pl (K.B.-S.); marta.surman@usdk.pl (M.S.)

**Keywords:** blinatumomab, acute lymphoblastic leukemia, immunotherapy, minimal residual disease, children

## Abstract

**Simple Summary:**

Immunotherapies are modern treatment modalities, giving hope for improvements of frozen cure rates in many childhood malignancies. More intensive cytotoxic chemotherapy cycles didn’t improve cure rates, only increase number of adverse events. Blinatumomab, a bispecific CD3/CD19 antibody construct, has been successfully used in relapsed/refractory r/r B-cell precursor ALL (BCP-ALL) as a bridge to hematopoietic stem cell transplantation (HSCT). We retrospectively assessed the efficacy and toxicity of blinatumomab in 13 children with r/r BCP-ALL. The response rate in our cohort of patients was 85%, with subsequent feasible HSCT in 11 out of 13 children. Although our study had some limitations with regard to its retrospective design and limited patient population, it clearly showed blinatumomab as not only a feasible but also an effective therapeutic option in pretreated children with r/r BCP-ALL, with a tolerable toxicity profile, paving the way for an HSCT procedure. To date, this is the first retrospective study from Poland on efficacy and toxicity of blinatumomab therapy in children with r/r ALL.

**Abstract:**

Despite the progress that has been made in recent decades in the treatment of pediatric acute leukemias, e.g., converting acute lymphoblastic leukemia (ALL) from a fatal to a highly curable disease, 15–20% of children still relapse. Blinatumomab, a bispecific CD3/CD19 antibody construct, has been successfully used in relapsed/refractory r/r B-cell precursor ALL (BCP-ALL) as a bridge to hematopoietic stem cell transplantation (HSCT). We retrospectively assessed the efficacy and toxicity of blinatumomab in 13 children with r/r BCP-ALL. Between 2017 and 2021, thirteen children, aged 1–18 years, with r/r BCP-ALL were treated with blinatumomab. Two patients were administered blinatumomab for refractory relapse without complete remission (CR), one due to primary refractory disease, and ten patients were in CR with minimal residual disease (MRD) ≥ 10^−3^. The response rate in our cohort of patients was 85%, with subsequent feasible HSCT in 11 out of 13 children. Ten children reached MRD negativity after the first blinatumomab administration. The three-year OS for the study patients was 85% (Mantel–Cox, *p* < 0.001) and median follow-up was 24.5 (range: 1–47). All responders proceeded to HSCT and are alive in CR, and MRD negative. Although our study had some limitations with regard to its retrospective design and limited patient population, it clearly showed blinatumomab as not only a feasible but also an effective therapeutic option in pretreated children with r/r BCP-ALL, with a tolerable toxicity profile, paving the way for an HSCT procedure.

## 1. Introduction

Acute lymphoblastic leukemia (ALL) is one of the most common pediatric malignancies, although childhood cancers are rare [1]. Despite the currently high complete remission rate (more than 95% newly diagnosed ALL patients) using risk-adapted protocols, approximately 15–20% of children will eventually relapse, with 8–10% dying from disease progression or treatment-related complications [2,3]. Survival rates after relapse significantly depend on leukemic blast immunophenotype (B-ALL vs. T-ALL) and time of relapse. The cure rates range from less than 30% for early relapses to 50−60% for late relapses [4].

Current standard-of-care therapies for relapsed/refractory ALL (r/r ALL), based on cytotoxic chemotherapeutic agents, are associated with severe acute and long-term toxicities, and quite often cause treatment-related death. New treatment modalities of reduced toxicity are an attractive option for heavily pretreated children and provide an opportunity to improve outcomes in those with r/r ALL. 

Blinatumomab, also known as BiTE, a bispecific T-cell engager, utilizes the patient’s own cytotoxic T cells to attack and induce lysis of CD19-expressing leukemic cells. Blinatumomab targets, and activates, two different antigen-binding sites: one directed toward tumor antigen CD19, expressed on leukemic blasts, and the other against receptor CD3 on T cytotoxic cells [5,6,7,8].

Many studies have confirmed the high effectiveness of blinatumomab in inducing complete remission in r/r B-cell precursor ALL (BCP-ALL) patients, both adults and children. Response rates range from 34% to 66%. Moreover, blinatumomab has an acceptable toxicity profile and is better tolerated than conventional chemotherapy [4,9,10,11,12,13]. Blinatumomab is also successfully used for the reduction of minimal residual disease in adults and children [3,8,13,14]. A significant molecular response was demonstrated by Gokbuget et al. in MRD-positive (≥10^−3^) adults in morphological remission. Among 113 evaluated patients, 88 (78%) achieved a complete MRD response. [15] Moreover, growing clinical data suggest that blinatumomab could be successfully used as a bridge therapy to allogeneic hematopoietic stem cell transplantation (HSCT) for r/r BCP-ALL patients [8,16]. The RIALTO trial of pediatric patients with CD19-positive r/r BCP-ALL demonstrated that children who reached complete remission after at least two blinatumomab cycles and proceeded to HSCT showed improved outcomes when compared to those who did not receive blinatumomab [8].

In this report, we present data on children with r/r BCP-ALL treated with blinatumomab in a compassionate, off-label setting as an effective bridging therapy to HSCT.

## 2. Materials and Methods

### 2.1. Study Design

We retrospectively collected data from the electronic medical records of children with r/r BCP-ALL who were treated with blinatumomab between 2017 and 2021 at the Department of Pediatric Oncology and Hematology, University Children’s Hospital of Krakow, Poland. 

Blinatumomab was administered according to ethical codes and regulations, with the approval of the Local Ethical Committee. Patients in our cohort were treated with blinatumomab on a compassionate, off-label basis. Since 2018, blinatumomab has been registered in Poland in monotherapy for the treatment of children aged 1 year or older with refractory or relapsed Philadelphia chromosome-negative, CD19-positive BCP-ALL, after receiving at least two prior therapies or after receiving prior allogeneic hematopoietic stem cell transplantation [17].

Hence, most of our patients were in second or further complete remissions (CR) at the time of blinatumomab administration with minimal residual disease (MRD) ≥ 0.1%. We obtained written informed consent from legal guardians of patients for off-label blinatumomab administration. 

### 2.2. Study Procedures

The bone marrow (BM) was evaluated before the first blinatumomab cycle and at day 29 of each cycle (the day after the termination of continuous infusion). 

Cytomorphological CR was defined as less than 5% of leukemic cells in the bone marrow. MRD assessment was performed using 10-color flow cytometry (MRD-FC). Each tube contained a ten-color panel: CD58-FITC/CD34-PE/CD45PerCP/CD10-PC7/CD19-APC/CD38-AF700/CD20-APC-H7/Syto-41/CD11a-BV510/CD117-BV605.

Cells were acquired using a FASCanto flow cytometer (Becton Dickinson, NJ, USA), and analyzed using FACSDiva version 8.0.1 software. 

A reduction of at least one-log fold in MRD load during blinatumomab treatment was defined as an MRD response. Children who presented a reduction in leukemic blast count from the initial number to less than 5% or a drop of at least one-log fold in blast count were considered responders, while the others were considered nonresponders. MRD levels < 0.01% (<10^−4^) were defined as negative. The expression of CD19 on leukemic cells was checked using flow cytometry upon diagnosis and each time bone marrow was evaluated during blinatumomab therapy. 

Children who achieved MRD negativity (<0.01%; <10^−4^) proceeded to HSCT. 

In the study, we also measured lymphocyte count; T-cell and B-cells were assessed before onset and at days 14 and 18 of each blinatumomab infusion. 

Adverse events were collected and assessed according to the Common Terminology Criteria for Adverse Events (version 4.0, 21 November 2021) [18].

### 2.3. Blinatumomab Dosage and Administration

All children were given blinatumomab in monotherapy as a 28-day continuous intravenous infusion through central line catheter, in an in-patient setting, with a 2-week, treatment-free interval after each cycle. Children who completed at least one blinatumomab cycle were included in the analysis. An initial dose of 5 mcg/m^2^ was escalated after one week to 15 mcg/m^2^/day. Since 2020, 15 mcg/m^2^/day has been used as an initial dose for children with a low tumor burden (MRD: 10^−2^–10^−4^; 1–0.01%), which included two patients in our cohort. A lower initial dose was indicated for patients with a high leukemic burden (BM > 25% leukemic blasts). During the screening, before the first blinatumomab cycle, and on day 29 of every cycle, each patient received a mandatory intrathecal central nervous system (CNS) infiltration prophylaxis with methotrexate, cytarabine, and prednisolone, with doses adjusted depending on age. To reduce the risk of cytokine release syndrome (CRS), steroid premedication was given. One dose of dexamethasone (10 mg/m^2^) was administered 12 h before the onset of blinatumomab infusion, and 5 mg/m^2^ at least one hour before infusion. 

### 2.4. Statistical Analysis

Due to the low number of patients, only a basic analysis was feasible. The main endpoint was overall survival (OS). OS was defined as the time from the date of starting blinatumomab infusion to death of any cause. If no event occurred, the observation was censored at the last follow-up. The date of the last follow-up was 15 October 2021. The Kaplan–Meier method was used to estimate survival probabilities, and differences between groups were compared by log-rank test. The significance level of 0.05 was used in all the statistical tests. Statistical analysis was performed by using SPSS (Statistical Package for the Social Science) version 25.0., 21 November 2021.

## 3. Results

### 3.1. Patients’ Characteristics

Between 15 December 2017 and 15 October 2021, thirteen children aged 1–18 years with relapsed/refractory BCP-ALL, Philadelphia chromosome-negative, were treated with blinatumomab at the Department of Pediatric Oncology and Hematology, University Children’s Hospital of Krakow, Poland. Children with r/r BCP-ALL presenting CD19 expression on leukemic cells were eligible for the assessment. 

Among the 13 children, eight were boys and five were girls. The median age at first diagnosis was 5 years (range: 8 months to 10 years). Almost all patients (85%) were treated according to ALL IC-BFM 2009 upon first ALL diagnosis. One child, initially diagnosed as an infant, was treated according to Interfant-06 protocol, and one child with third BCP-ALL relapse was treated according to ALL-IC-BFM2002 for his first disease. Ten out of 13 (77%) were stratified to the intermediate-risk group (IR), and three (23%) to the high-risk group (HR). The majority of children (12 of 13) in a relapse situation were treated according to IntReALL 2010 Protocol (I-BFM-SG International Study for Treatment of Childhood Relapsed ALL). Initially, six children were classified as standard-risk (SR), based on a late relapse; however, due to an unsatisfactory response to therapy, shown as MRD ≥ 10^−3^, measured at the end of induction, they were upgraded eventually to HR. The median time since the beginning of chemotherapy for relapse/refractory BCP-ALL and the implementation of blinatumomab was 3.3 months (2.4–9 months).

The median age at the onset of blinatumomab therapy was 8 years (range: 1 to 17 years). 

Patients had Karnofsky or Lansky (age < 16 years) performance status of ≥50%. Patients’ characteristics are shown in detail in Table 1.

Two patients (15%) were administered blinatumomab for refractory relapse (BM > 25% blasts), one due to primary refractory disease, and ten patients (77%) were in the second or further cytomorphological CR with positive MRD (≥10^−3^). 

There was no patient who relapsed after HSCT. Among the twelve relapsed children, 10 presented with a first relapse, and two with a second. The median time from diagnosis to relapse was over 4 years (50 months; range: 6 months to 10 years). One patient presented with very early relapse (less than 18 months from diagnosis), two with early relapse (between 18 and 36 months from diagnosis), and nine with late (more than 36 months since diagnosis) first or further relapse. The majority of patients presented isolated bone marrow relapse (11 children), whereas two patients had a combined relapse (BM/CNS and BM/testes). The patient with combined BM/CNS relapse presented with facial nerve paresis of both sites with no signs of CNS involvement in imaging studies before blinatumomab therapy. Most of the patients (77%) were given one salvage therapy protocol after relapse (one induction block and one or two consolidation chemotherapy blocks). 

### 3.2. The Number of Blinatumomab Cycles and Treatment Modifications

Our cohort of pediatric patients received 17 complete blinatumomab cycles. Seven patients completed one cycle of blinatumomab, while five patients received two cycles. Due to neurotoxicity grade 4 CTCAE, one patient discontinued treatment after 4 days infusion of the first blinatumomab cycle. This child was eligible for analysis (ITT, intention-to-treat approach). Two patients who presented neurotoxicity grade 3 CTCAE (seizures and/or tremor) continued blinatumomab infusion with dexamethasone, dosed at 5 mg/m^2^.

### 3.3. Outcomes

In our study, 10/12 (83.3%) children who received at least one complete blinatumomab cycle responded to blinatumomab therapy. The child whose blinatumomab infusion was stopped due to grade 4 neurotoxicity reached MRD < 0.01% after 4 days infusion and was transplanted. Thus, the overall response rate in our cohort of patients (13) to blinatumomab was 85%. 

Nine of 12 children (75%) achieved MRD negativity (<0.01%) after the first blinatumomab cycle, with prior cytomorphological CR. The mean disease load before the first blinatumomab cycle was 30.6% (range: 0.01–55%), with a reduction to 11.3% (range: 0–98%) after the first blinatumomab administration. Among the three patients who presented with high leukemic load (>25%) at the onset of blinatumomab therapy, only one responded to blinatumomab, with a leukemic load reduction from 32.7% to 0.36% after one cycle and reaching MRD negativity after the second. The other two children presented 37% and 55% leukemic blasts in the bone marrow before blinatumomab administration, and 28% and 97% blasts after the first cycle, respectively, and were considered as nonresponders. The patient with disease progression was disqualified from any further therapy and transferred to palliative care. The other nonresponder (KMT2A rearrangements, first diagnosis at 8 months, primary refractory disease) received further salvage chemotherapy, with no response, and died. The immunophenotype of blasts assessed in this patient showed negativity of CD19 after BiTE (blasts immunophenotype before BiTE administration: CD19bright/CD45med/CD10neg/CD34pos/CD33pos/NG2dim, and after BiTE: CD19negCD45med/CD10neg/CD34pos/CD33pos/NG2dim).

The blinatumomab approach was consolidated by HSCT in 11 out of 13 children (85%). Two children received HSCT from haploidentical donors, three from matched related donors, and six from a matched unrelated donor. 

Of note, in our pediatric cohort, no relapse was observed after blinatumomab with subsequent HSCT. The median follow-up was 25.4 months (range: 1–47 months) for the whole study group. Responders to blinatumomab therapy (at least one-log reduction of initial MRD load) presented 100% OS, while lack of response resulted in poor outcome (2/2 deaths) (Figure 1). The three-year OS for the whole group was 85% (Mantel–Cox; *p* < 0.001). To date, 11 patients treated with blinatumomab are alive, after HSCT, in continuous remission, and negative for MRD-FC. 

### 3.4. Toxicity of Blinatumomab Therapy 

Blinatumomab was generally well-tolerated, and toxicity was easily manageable in the majority of patients. More than half of the patients (54%; 7/13) presented adverse events of grade 3 CTCAE. Among adverse events of blinatumomab therapy, hematological toxicities were the most common, mainly anemia and thrombocytopenia, aligning to grade 3 and 4 CTCAE (Table 2). However, in the majority of children, cytopenia was observed before the first administration of blinatumomab and was thus caused by earlier cytotoxic chemotherapy cycles. 

Fever (38%) and headache (30%) were also often reported. One child developed a reversible neurotoxicity (grade 4 CTAE), with coordination, balance, and speech disturbances, as well as multifocal seizures presented on the fourth day of the first blinatumomab infusion. Since the patient had a low MRD load (0.01%) before blinatumomab administration, and BM after blinatumomab cessation showed MRD negativity, the therapy was discontinued, and the patient proceeded to HSCT (Patient 4, Table 3). One patient with high initial leukemic load (37%) developed reversible cytokine release syndrome (CRS, grade 3 CTCAE) on the first day of blinatumomab administration. Five days later, after stabilization of the patient’s general status, the blinatumomab infusion was restarted and continued with dexamethasone support. The second blinatumomab cycle was administered without concomitant steroids. The patient has been alive for 47 months since the start of blinatumomab treatment (Patient 1, Table 3) Two patients had a few hours break in blinatumomab infusion, one due to neurological symptoms (seizures, grade 3 CTCAE) and one due to reversible tachycardia with narrow QRS. The latter symptom resolved spontaneously. Two children with low MRD before blinatumomab introduction, who did not follow the step-up regimen and received a higher initial BiTE dose (15 mcg/m^2^/day), presented fever and headache during the first days of BiTE administration (grade 2 CTCAE). These side effects were easily manageable with antipyretic medications (paracetamol). There were no fatal adverse events in the study. Adverse events that occurred in patients are presented in Table 2. 

### 3.5. T-Cell and B-Cell Kinetics

The majority of patients presented low counts in both B- and T-cells. At baseline, the mean value for B-cells was 70 cells/µL (range: 10–110). The B-cell count remained below the detection limit for the entire treatment period (data not shown). The mean T-cell count at baseline was 451/µL (range: 200–865). After a transient drop of peripheral T-cells at the end of the first cycle (mean: 252/µL), a subsequent increase (mean: 865/µL) after the second cycle was seen. Responders had a more pronounced increase in T-cells than nonresponding patients; however, due to the small number of patients, statistical analysis was not performed. Figure 2 shows the T-cell kinetics during the first blinatumomab cycle.

## 4. Discussion

Despite enormous progress being made in recent decades in the treatment of pediatric acute leukemias, e.g., converting ALL from a fatal to highly curable disease, relapsed/refractory patients still have a dismal prognosis [2,19]. Increasing the intensity of chemotherapy significantly increases the number of adverse events, rather than improving treatment outcomes. Modern therapies, such as T-cell-based therapeutic strategies, present new therapeutic options of strong efficacy [3,13,20].

Our retrospective, single-center analysis of blinatumomab therapy in children with refractory and relapsed BCP-ALL confirms efficacy in reducing leukemic/disease load and reaching cytomorphologic and cytometric remission before approaching an HSCT procedure. The response rate in our cohort of patients to blinatumomab was 85%, with subsequent feasible HSCT in 11 out of 13 children; however, there were a limited number of patients in our study.

Blinatumomab efficacy in the pediatric population was demonstrated in many clinical trials. von Stackelberg et al. reported data on 70 children, with a CR rate of 39% (27/70) after one to two cycles of blinatumomab [9]. The recently published study by Quedueville et al. showed a response rate of 34% (13/38) [10]. However, in both studies, the majority of patients received blinatumomab infusion for relapse with excessive blast infiltration of bone marrow (74% of children presented above 50% of blasts in bone marrow, and 71% of children presented above 25% of blasts, respectively) [9,10]. The high response rates in our study cohort compared to other published data could be associated with the lower leukemic load before blinatumomab therapy in most of our patients. The mean disease load before the first blinatumomab infusion in our cohort of patients was 30.6% (range: 0.01–55%); however, the majority of the children (10/13) had an MRD level ≤ 0.2%. Our data suggest that a lower MRD load before blinatumomab administration is strongly correlated with superior response and outcome, as previously described [9,10,11,21]. In the RIALTO trial, an open-label, single-arm international study of pediatric patients with CD19-positive r/r BCP-ALL, one of the largest pediatric cohorts (110 patients), most of the patients (n = 98) had ≥5% leukemic blasts in bone marrow at baseline. After two blinatumomab cycles, 59% of patients achieved CR. Out of 12 patients with <5% of leukemic blasts at baseline, 92% achieved an MRD response. The patients who achieved an MRD response had longer OS than those who achieved CR without an MRD response (median OS: 21.2 months vs. 14.1, respectively) [8].

Nevertheless, our data with high response and survival rates are quite unique among other studies [9,10,11]. Apart from the low tumor load before blinatumomab infusion, and CR with MRD positivity in the majority of our patients, other factors should also be considered. Most of our patients presented with late relapse (10/13), with a median of 50 months from the first diagnosis to relapse. Genetic abnormalities of known poor prognostic impact were found in only two patients (KMT2A/MLL; hypodiploidy) (Table 1). Moreover, most of the patients were recruited to blinatumomab therapy at an early stage of salvage relapse therapy. The median time from beginning chemotherapy for relapse/refractory BCP-ALL to the implementation of blinatumomab was rather short, 3.3 months (2.4–9 months). No deaths after HSCT were reported in our cohort. The majority of our patients did not present serious adverse events either before or during blinatumomab implementation; thus, they approached the HSCT procedure in good general status and condition (Karnofsky or Lansky performance scale of 70–90%), affecting the outcome. One should underline the retrospective design and the limited number of patients in our study; thus, the final high response rate should be taken with caution.

Locatelli et al. showed that HSCT improves the outcomes in r/r BCP-ALL patients previously treated with blinatumomab [8], which was also shown by our data. However, the low number of patients in our cohort and the lack of a comparable study arm with children treated with blinatumomab not bridging to HSCT are considerable limitations of the presented study.

In another recent trial conducted by Locatelli and others, blinatumomab was successfully used instead of a third consolidation in high-risk, first-relapse BCP-ALL children. MRD response was observed in more patients in the blinatumomab group vs. consolidation chemotherapy group, 90% (44/49) vs. 54% (26/48); thus, more patients from the blinatumomab group were able to undergo HSCT. The randomization was stopped earlier, due to superiority of outcome and lower incidence of adverse events in the blinatumomab group (31% vs. 57%, log-rank *p* < 0.001) [22]. Currently, blinatumomab is under investigation in randomized pediatric studies not only in relapsed patients but also as front-line therapy for some subsets of high-risk patients [23,24,25,26].

Despite earlier treatment with chemotherapy in the majority of our patients, the toxicity rates and profile during BiTE administration were acceptable. Adverse events in our study were comparable to those reported by others with respect to frequency and severity [8,11]. More than half of the patients (7/13) experienced grade 3 adverse events, although most of these were preexisting hematological toxicities. None of the study patients presented a fatal adverse event. Neurologic adverse events resulted in a temporary interruption of blinatumomab treatment. There were only 4/13 (30%) severe adverse events, with one reversible CRS presentation, grade 3 CTCAE, in a patient with high leukemic burden at the initiation of blinatumomab; the patient responded to therapy, achieved CR after one cycle of blinatumomab, and a negative level of MRD after the second. In previous studies, most responding patients presented some degree of CRS; however, there seems to be no direct association between the severity of CRS and response to therapy [8,10]. As already shown, the leukemia load before blinatumomab therapy is associated with the risk of CRS [27]. Since most of our patients had a leukemic load at a baseline of <5%, we did not observe CRS, except in one patient.

In some patients with low MRD loads, we resigned from the step-up dosing regimen in the first cycle and started with a higher dose (15 mcg/m^2^/day), as implemented in ongoing studies for patients with persistent MRD [28]. Apart from a higher incidence of fever and headache in the first days of BiTE administration, we did not observe any further toxicities.

According to the product characteristics for blinatumomab, for pediatric patients without CR, with a high initial tumor load, hospitalization is recommended for the first 9 days of the first cycle and the first 2 days of further cycles of blinatumomab continuous infusion. For patients in CR with persistent MRD, hospitalization is recommended for 3 days of the first cycle and 2 days of the second cycle. Due to technical reasons, patients were hospitalized uninterrupted during the time of blinatumomab infusion. Nevertheless, tolerance of the therapy and quality of life of our cohort during blinatumomab treatment were superior compared to that experienced during conventional chemotherapy, which was also reported by other groups [29,30].

In our study, we reported T-cell expansion in responders compared to nonresponders (Figure 2). This is in line with the finding of Zugmeier et al., who reported T- and B-cell kinetics in adult patients with relapsed/refractory BCP-ALL during and after blinatumomab therapy. It was shown that long-term survivors (OS ≥ 30 months) had more pronounced T-cell expansion and peripheral B-cell depletion as compared to minor or even absent T-cell expansion in those with OS < 30 months [31]. A different observation was made by Queudeville et al. in a pediatric setting, with no difference in the initial absolute count of T-cell subsets between the responding and nonresponding patients [10]. The two nonresponder patients in our study were heavily pretreated and had a low number of T-cells at the onset of blinatumomab therapy, which could be partly the reason for treatment failure. However, no firm conclusion can be drawn because of the low number of patients (10 responders vs. two nonresponders).

Although there are data available suggesting that dexamethasone administration does not have a negative impact on cure rates of blinatumomab recipients, we tried to omit steroid prophylaxis in further blinatumomab cycles due to possible impairment of T-cell function [32].

Many questions are yet to be answered with regard to the optimal timing of administration and number of cycles, and management of CD19-negative relapses. However, the future is open for a blinatumomab approach, since the efficacy and safety profile in clinical trials are outstanding.

## 5. Conclusions

In conclusion, we presented outcomes of blinatumomab therapy in children with relapsed/refractory ALL treated in a compassionate, off-label setting before HSCT, planned as the last phase of salvage therapy. Blinatumomab is a well-tolerated agent that can induce MRD remission in pediatric patients with relapsed BCP-ALL and can be used as a bridging therapy to facilitate subsequent HSCT.

## Figures and Tables

**Figure 1 cancers-14-00458-f001:**
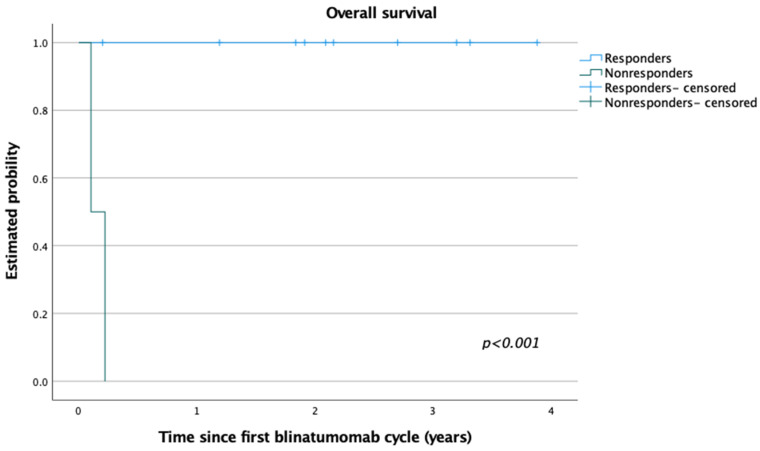
Overall survival (OS) for patients who responded to blinatumomab therapy (responders) vs. those who did not respond to the first cycle (nonresponders). Three-year OS for the study patients = 85% (Mantel–Cox, *p* < 0.001; median follow-up, 25.4 months).

**Figure 2 cancers-14-00458-f002:**
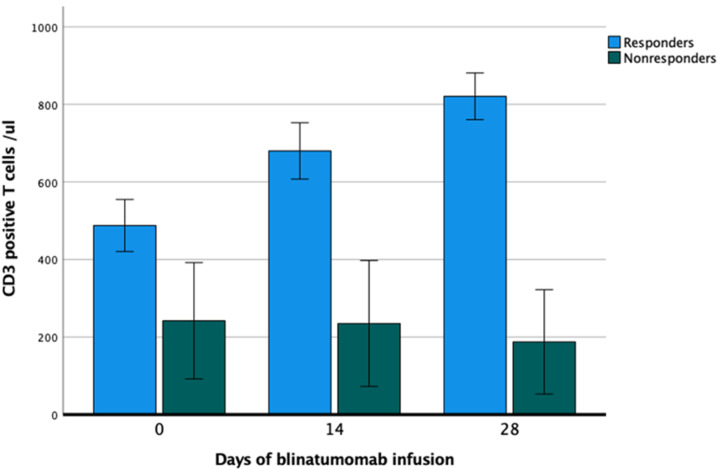
T-cell kinetics during first blinatumomab cycle (days 0, 14, and 28).

**Table 1 cancers-14-00458-t001:** Patient’s characteristics.

Features	
**Age at initial diagnosis in years**	
Median (range)	5.0 (8 months–10 years)
**Age at onset of blinatumomab**	
Median (range)	8.0 (1–17 years)
**Sex (n (%))**	
Boys	8 (61.5%)
Girls	5 (38.5%)
**Genetic aberration**	
ETV6-RUNX1	2
KMT2A	1
Hyperdiploidy	3
Hypodiploidy	1
No known genetic aberration	6
**Disease status prior blinatumomab therapy**	
Refractory disease	1
1st relapse	10
2nd relapse	1
3rd relapse	1
**Time of relapse**	
Very early (<18 months from diagnosis)	1
Early (>18 and <36 months from diagnosis)	2
Late (>36 months from diagnosis)	10
**Leukemia load at onset of blinatumomab therapy (leukemic blasts in BM (%))**	
>50	1
25–50	2
5–25	0
<5	10
**Extramedullary manifestation before onset of blinatumomab (n)**	
CNS * (facial nerves paralysis/infiltration in MRI)	1

BM—bone marrow; CNS—central nervous system. * Before onset of blinatumomab, patient presented resolution of leukemic infiltration in MRI image.

**Table 2 cancers-14-00458-t002:** Toxicity of blinatumomab therapy observed in study patients.

Toxicity of Blinatumomab	Number of Patients (Total = 13)
**All adverse events (AEs), any grade**	20
grade 3 CTCAE	7
grade 4 CTCAE	3
**Severe adverse events (SAEs)**	4
AE leading to treatment interruption	3
AE leading to treatment cessation	1
Fatal AE	0
Cytokine release syndrome grade 3 CTCAE	1
Anemia	8
Thrombocytopenia	4
Neutropenia	5
Hypotension	1
Seizures	2
Tremor	2
Headache	4
Fever	5
Increase of AST/ALT grade 3 CTCAE grade 4 CTCAE	4 1
Hypertension	1
Legs pain	2
Hypokalemia	3
Hyperferritinemia	4

AST—aspartate transaminases; ALT—alanine transaminase.

**Table 3 cancers-14-00458-t003:** Blinatumomab treatment outcome and MRD response.

Number N = 13	MRD-FC Prior 1st Cycle	MRD-FC Post 1st Cycle	MRD-FC Post 2nd Cycle	Responder (R) vs. Nonresponder (NR)	Treatment Post Blinatumomab	Follow-Up Duration (Months)
1	32.7%	0.36%	<0.01%	R	MUD HSCT	47
2	0.1%	<0.01%		R	MSD HSCT	40
3	0.1%	<0.01%		R	MSD HSCT	38
4 *	0.01%	<0.01%	-	R	MUD HSCT	39
5	55%	97%		NR	Palliative care	1 DEAD
6	0.03%	<0.01%	<0.01%	R	MUD HSCT	32
7	37%	28%		NR	Clofarabine	3 DEAD
8	0.03%	<0.01%	<0.01%	R	MSD HSCT	26
9	0.2%	<0.01%		R	haploidentical HSCT	23
10	0.15%	<0.01%	<0.01%	R	haploidentical HSCT	22
11	0.01%	<0.01%	<0.01%	R	MUD HSCT	14
12	0.2%	<0.01%		R	MUD HSCT	25
13	0.15%	<0.01%		R	MUD HSCT	2

MSD, matched sibling donor; MUD, matched unrelated donor; HSCT, hematopoietic stem cell transplantation; R, responder to blinatumomab; NR, nonresponder to blinatumomab. * Patient 4 discontinued blinatumomab treatment after 4 days.

## Data Availability

The data presented in this study are available upon reqest frem the corresponding authors. The data are not publicly available due to privacy and ethical restrictions.

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
