# Peer review of "Blinatumomab as a Bridge Therapy for Hematopoietic Stem Cell Transplantation in Pediatric Refractory/Relapsed Acute Lymphoblastic Leukemia"

_cancers, 2022, doi:10.3390/cancers14020458_

Round 1

Reviewer 1 Report

Pawinska-Wasikowska and others describe outcomes of a cohort of children with relapsed/refractory B-cell ALL who received blinatumomab, the majority of whom were treated in morphologic remission and persistent MRD. They show very nice outcomes for the patients who responded to this therapy, but this paper would likely benefit from more emphasis on what makes this relatively small cohort unique.

Major Comments:

  1. Discussion (pages 5-6): Instead of repeating the data that are stated primarily in the Results section, can you please try to highlight in what ways the results are unique relative to other, larger series? For example, are there any explanations for why not a single death was seen among the responders, even though they all underwent HSCT? How does this compare to patients who experience an excellent response to traditional chemotherapy at your center? The paragraph describing the T-cell kinetics does a nice job explaining how the data can be put into context with other studies.
  2. Discussion (page 6): Unless I missed it, this is the first mention of the strategy of elimination the “step-up dosing regimen.” If I am correct, this should be mentioned in the Methods first, along with more details about the toxicities observed in these patients in the appropriate section of the Results.

Minor Comments:

  1. Introduction (page 2): Blinatumomab is not a “bispecific monoclonal antibody.” For example, it lacks an Fc region that a full antibody molecule would possess. Please refer to it specifically as a bispecific T-cell engager.
  2. Results, Outcomes (page 4): It is not really accurate to refer to someone with >5% blasts as having MRD. Instead of the term “MRD load,” consider something more broad like “relative disease burden.”

Author Response

Thank you for all remarks. The article has been revised according your suggestions. I hope it is more clear now, and relatively good outcomes reached by study patients, i.e. no deaths after HSCT has been even partially explained. 

Reviewer 2 Report

The paper focuses on a single-center experience with blinatumomab treatment in pediatric r/r ALL as a bridge for HSCT setting that is of moderate interest for pediatric hematologist and offers some moderate insight into the field. The studied group is relatively small (data on several-times larger cohorts are available now) and the indications for treatment are not new - there are several reports on the off-label use of blinatumomab as a bridge for HSCT in children (as well as adults), and therefore the paper lacks novelty. 

Major issues

  • lack of novelty, retrospective nature, and small sample size make the paper of moderate interest to the readers and as such should be considered for publication in lower-range impact journal
  • the authors decided to exclude the data of grade 4 toxicity patient from the analysis - I believe the data from intention-to-treat analysis would be of greater benefit to the readers
  • genetic background information is very basic and therefore does not add much to the existing body of data; similarly no in-depth characteristics analysis of patients with high-grade toxicity was included
  • B- and T-cell kinetics data is very basic and does not add much to the existing body of data. Moreover, in section 4.2 of the results (B and T-cell kinetics) the data provided in the text is not reflected in the figure (Fig.2 shows only T-cell counts during the first cycle, while the kinetics between the first and the second cycle are discussed in the manuscript). 
  • discussion is quite short and does not include all the available clinical data on the topic. The available data on the use of blina as a bridge for HSCT were not included in the introductory or discussion part, therefore suggesting that this is the first paper reporting such indication.
  • Kaplan-Maier curve should include information on follow-up duration at the moment of data-cut-off for all surviving patients as it may be misleading in the present form. 

Minor issues:

  • the division of M&Ms section into subsections is unclear and misleading; most of the methods has been written under the first subsection ("methods") and it is unclear why the others (e.g. 2.3. T- a B-cell kinetics) were given a separate subsection. I'd suggest repharsing, to make sure that the text divisions into subsections reflect the content and are logically justified.
  • in the discussion the authors refer to the quality of life of the cohort (discussion, paragraph 7) without data supporting that claim (QoL was not assessed in any unbias way)
  • "outcome" column in table 2 is not very informative; I'd rather see the survival time/follow-up duration + info on deaths in respective cases
  • there are a few typos, punctuation errors, and minor language/grammar issues to be corrected at proofreading stage (e.g. some "international" words were not translated into English forms - hiperdiploidia, clofarabina (...), lack of plural forms when needed, issues with article use etc.)

Author Response

Thank you for all remarks. The article has been revised according your suggestions. For sure the main limitation of the study is small sample size, as well as lack of comparable study arm (patients treated with blinatumomab who did't proceed to HSCT).

Although there are many studies on blinatumomab use as a bridge to HSCT in relapsed/refractory B-ALL children, we decided to report it anyway mainly due to satisfactory outcomes reached in our cohort, and acceptable toxicity incidence and profile. Discussion has been revised according to your suggestions.

According to your suggestions ITT analysis was done, and results have been revised. 

We didn't show B-cells kinetics in our patients mainly because its count was extremelly low and constant during the entire blinatumomab therapy in patients.

Kaplan-Maier curve and survival data have been also revised; and information about duration of follow up was presented.

Thank you again for all remarks. Hope the paper is more clear and interesting now.

Reviewer 3 Report

PawiÅ„ska-WÄ…sikowska et al. describe the use of Blinatumomab (BLIN) in R/R ALL in pediatric patients. The subject is clinically relevant as Blin is known to improve outcomes in R/R ALL. Here are my comments.  

Major comments:

BLIN is now indicated as first-line therapy for MRD +ve ALL in patients with CR after induction therapy and not just in R/R setting. Studies have shown that MRD negativity after BLIN use followed by HSCT decreases relapse and improves outcomes in ALL. The authors should discuss this.

Total 10 patients responded to BLIN, and all of them underwent HSCT. None of those patients relapsed. MRD negativity with BLIN helped with the outcomes, but it should also be discussed that HSCT was another major factor in improving outcomes.

Limitations of the study e.g. retrospective design, low patient population should be discussed.

In the discussion section, the authors mention that they aimed to find the optimal steroid dosing with BLIN, but they did not mention the optimal dosing they found. I would suggest either giving more details or deleting that statement altogether.

The authors mention that IC-BFM was chemotherapy at diagnosis. What other chemotherapy was used in relapsed patients before BLIN?

Authors conclude that improved outcomes in their results were likely due to low disease burden in the patients included in the analysis, compared to high disease burden in other studies. Mention the average residual disease in the studies by von Stackelberg and Quedeuville et al. before Blin use to compare with the MRD load in this study.

Minor comments

The manuscript should be edited for English language and grammatical errors.

Use of reference in the middle or at the end of the sentence should be uniform.

Define BCP in the main manuscript and not just on abstract.

Author Response

Thank you for all remarks. The article has been revised according your suggestions.

New indications for blinatumomab use, apart from presented in paper, r/r ALL in CR, MRD+ and noCR, i.e front line use were mentioned.

We tried to present our data clearer, to show the significant HSCT role  in the final outcome. Thank you for underlining this issue.    

For sure the main limitation of the study is small sample size, retrospective designas well as lack of comparable study arm (patients treated with blinatumomab who did't proceed to HSCT). These issues have been discussed. 

The data about optimal steroid dose were removed; it concerned one patient.

Information about chemotherapy in relapse before blinatumomab were given.  

Thank you again for all remarks. Hope the paper is more clear and interesting now.

Round 2

Reviewer 1 Report

The authors appear to have addressed my requested comments sufficiently.

Author Response

We attached the revised paper.

Reviewer 2 Report

The authors have revised the manuscript according to the suggestions, however, as mentioned in the first review, the major issues and limitations that result from the study design could not be changed/improved. I believe the manuscript could be published in the present form if it is considered in scope/in accordance with the journal editorial policy.

Author Response

Thank you for your comment. We followed your suggestions.

This manuscript is a resubmission of an earlier submission. The following is a list of the peer review reports and author responses from that submission.